# Studying the Bulk and Contour Ice Nucleation of Water Droplets via Quartz Crystal Microbalances

**DOI:** 10.3390/mi12040463

**Published:** 2021-04-20

**Authors:** Karekin Dikran Esmeryan, Nikolay Ivanov Stoimenov

**Affiliations:** 1Acoustoelectronics Laboratory, Georgi Nadjakov Institute of Solid State Physics, Bulgarian Academy of Sciences, 72, Tzarigradsko Chaussee Blvd., 1784 Sofia, Bulgaria; 2Department of Distributed Information and Control Systems, Institute of Information and Communication Technologies, Bulgarian Academy of Sciences, Acad. G. Bonchev Street, Bl.2, 1113 Sofia, Bulgaria; nikistoimenow@gmail.com

**Keywords:** freezing mode, outside-in freezing, quartz crystal microbalance, soot

## Abstract

Due to the stochastic and time-dependent character of the ice embryo formation and growth (i.e., a process that can be analyzed statistically, but cannot be predicted precisely), the heterogeneous ice nucleation on atmospheric aerosols or macroscopic solid surfaces is still shrouded in mystery, regardless of the extremely active research and exponential progress within this scientific field. For instance, whether the icing appears from outside-in or inside-out is a subject of intense controversy, with practicability in designing passive icephobic coatings or improving the effectiveness of the cryopreservation technologies. Here, we propose an artful technique for quantitative analysis of the different modes of water freezing using super-nonwettable soot-coated quartz crystal microbalances (QCMs). To achieve this goal, a set of 5 MHz QCMs are loaded one at a time with a 50 μL droplet, whose bulk or contour solidification is detected in real-time. The obtained experimental results show that our sensor devices recognize explicitly if the ice nuclei form predominantly at the liquid–solid interface or spread along the droplet’s entire outer shell by triggering individual reproducible responses in terms of the direction of signal evolution in time. Our results may serve as a foundation for the future incorporation of QCM devices in different freezing assays, where gaining information about the ice adhesion forces and ice layer’s thickness is mandatory.

## 1. Introduction

Apart from being pernicious to our hectic daily routine, by blocking the road infrastructure and the operability of the telecommunication towers, high-voltage power lines, wind turbines, aircraft, air conditioners etc. [1,2,3,4,5,6], the water’s phase transition into ice has more global implications of fundamental importance. For instance, the ambient water vapor may condense/freeze around soot aerosols available in the atmosphere due to anthropogenic activities, thus forming mixed-phase or cirrus clouds that strongly affect the Earth’s climate system (radiative balance) [7,8,9].

Of particular scientific interest appears to be the mechanism of ice nucleation around the individual soot particles and associated solid aggregates/agglomerates, and the decades of extensive research in this field reveal that the water uptake ability of the aerosols, their wettability and the porosity of the individual soot particles are dominating the freezing process [8,10,11,12,13]. It is found that the pore-free particles inhibit the formation of ice embryos [12], which is achievable also in the absence of a three-phase contact line i.e., when the particle is fully immersed in a liquid (e.g., in supercooled droplets occupying the clouds at high altitudes) [14].

Nevertheless, the role of the interfacial contact line in regulating the water freezing remains poorly understood and provokes scientific discussions with many open questions, mainly related to elucidating whether the droplet’s heterogeneous nucleation takes place from outside-in or inside-out [14,15,16,17,18]. Using the three-state Potts model and a computer simulation, orders of magnitude higher nucleation rate at the particle–liquid interface, compared to the case of a particle immersed in a water droplet, is clearly determined [14]. The rationale of this observation is that upon expansion of the ice nucleus at the three-phase contact line, the area of all three interfaces (solid–air, solid–liquid and liquid–air) decreases, triggering a corresponding reduction in the interfacial tensions and Gibbs thermodynamic free energy barrier [14]. Hence, an excellent experimental work demonstrates that the location of ice incipiency can be effectively shifted to the liquid–air interface if a nanotextured membrane is immersed in the droplet [16].

Nowadays, the initiation of freezing events at the droplet’s free surface holds very important practical applications in developing sustainable passive icephobic coatings, whose operation principle relies on the concentric inward motion of the freezing boundary from the contour (outer shell) of the droplet towards the slurry liquid bulk, inducing volumetric expansion and incompressibility of the droplet and its subsequent lifting (“self-dislodging”) from the solid surface [19]. Moreover, the evaporative cooling of millimetric water droplets placed on a soot coating causes spontaneous outside-in freezing and due to the generated internal pressure, the incompletely frozen droplets may explode [20].

Recently, our research group has accidently discovered a “hybrid version” of the outside-in freezing phenomenon, noticed by assessing the freezing temperature depression of water droplets resting on a variety of soot coatings (see the next sections). Wehave found that the simultaneous cooling of the coating and the liquid, in order to determine the droplet’s solidification temperature, leads to freezing of both the contact interface and the droplet’s entire outer shell. Meanwhile, when the soot surface is preliminary cooled (for evaluating the water’s freezing time delay, a term equivalent to nucleation time-lag), the aforementioned contour freezing mode is missing, and the droplet freezes in the conventional two-steps’ way: partial solidification at the contact line (recalescent freezing) followed by isothermal freezing of the remaining bulk liquid (referred to as a bulk mode). These findings validate the hypothesis of *Esmeryan* et al. that the cryopreservation of human spermatozoa via soot is successful due to outward osmosis, as a result of the contour freezing, and balanced dehydration of the cells [21]. Considering the vital interest of the atmospheric and materials scientists in completely clarifying the origin and physics of the various modes of water freezing [11,12,13,14,15,16,17,18,19,20], it would be highly beneficial to suggest a reliable sensor device capable of detecting and analyzing them.

Undoubtedly, the quartz crystal microbalance (QCM) is one of the appropriate piezoresonance sensors that can track any water freezing events in a highly sensitive and reproducible manner, since it reacts instantly to negligible changes in the mass added to its surface (absolute sensitivity of ng/cm^2^) [22] and to viscosity–density alterations in the liquids [23,24,25] or surface tension and wettability variations [26,27,28]. Furthermore, the QCM is provenly efficient in sensing the water’s phase transitions [29,30,31] and defining the anti-icing characteristics of superhydrophobic surfaces [32].

The primary objective of this study is to represent a new platform for accurate analysis and facilitated future interpretation of the bulk and contour ice nucleation modes occurring in water droplets through soot coated QCM sensors. We show, experimentally, that the proposed sensor configuration can unambiguously distinguish whether the inception of ice nuclei is solely at the solid–liquid interface or along the droplet’s entire shell, by generating recognizable resonance responses concerning the signal trend. A substantial superiority of our approach, compared to the commonly used high-speed video imaging, is the possibility of calculating the thickness of the ice layer formed throughout the recalescent freezing stage—a task that has not been implemented yet.

## 2. Materials and Methods

### 2.1. Soot Deposition

The soot used as an interfacial coating for a series of six 5 MHz QCMs (SRS, San Diego, CA, USA) is obtained during the controlled combustion of rapeseed oil at atmospheric pressure [31,33]. The pyrolysis inside a conical steel chimney induces the inception of pericondensed polyaromatic hydrocarbons, part of which do not interact (burn) with the surrounding oxygen, reaching the chimney through a flow meter/controller (Fisher Scientific 11998014, Schwerte, Germany), and cluster to create soot particles. These particles are deposited on the quartz crystals’ surface by subjecting the latter to the flame/fume emitted from the chimney’s tip for 15 s. The flame’s laminarity and the fluently circular movement of the specimens across the fume ensure uniform deposition of ~16 ± 1 μm thick soot coatings with a fractal-like structure, quasisquare aggregate/agglomerate morphology, root mean square roughness of ~100 nm, inherent mechanical durability and non-wettability towards water and other liquids, including human semen and urine [21,25,28], as shown in Figure 1.

### 2.2. QCM Based Sensing Platform for Analyzing the Dynamics of Ice Nucleation on Super-Nonwettable Carbon Soot Coatings

The occurrence of bulk and contour ice nucleation is traced using a QCM 200 analytical instrument (SRS, San Diego, CA, USA) consisting of a digital controller with a built-in frequency counter and ohmmeter, a sensor oscillator SRS 25 and a quartz crystal holder intended to accommodate the QCM device. The real-time sensor signal is continuously recorded on a personal computer at 1 s gate time by means of LabView Stand Alone computer software (SRS, San Diego, CA, USA).

Prior to starting the experiments, there was one technical drawback that had to be overcome. The operating thermal range of our measurement system is within 0 to +40 °C [34], while the icing assays must be performed at negative temperatures up to −30 °C. In addition, taking into account that the crystal holder is made of Teflon, the insulating properties of this material may create a substantial mismatch between the measured cooling temperature (see the next subsection) and the actual temperature of the soot coated QCM in a given timeframe, thus compromising the interpretation of the readings. Therefore, we decided to design and fabricate a unique thermally conductive copper-based QCM holder, illustrated in Figure 2.

The as-developed holder is composed of an upper and lower part connected to one another by two M4 screws. An appropriate nest, where the 1-inch quartz crystal should be mounted, is carved at the lower (bottom) part, whereas a technological clearance among both metal details is ensured when assembling them, with an aim of eliminating short circuits in the electrical branch of the sensor and its subsequent damage. Major advantages of our custom-designed holder, compared to the SRS’ and other companies’ counterparts, are the wide working thermal range (−200 °C to +1000 °C), the excellent thermal conductivity of the copper and the large contact surface minimizing the temperature gradients between the surrounding environment, the holder’s fuselage and the QCM sensor. The lack of such gradients is verified with a thermal imaging camera FLIR P640 (FLIR systems, Wilsonville, OR, USA) and the maximum observed temperature deviations are ±0.2 °C. Moreover, the length of the coaxial cable (see Figure 2) is identical to that of the original Teflon-made holder, which warrants continuous resonant oscillations up to acoustic (electromechanical) loads of 5000 Ω [34].

### 2.3. Icing Assays

The bulk and contour ice nucleation discerned while studying the freezing time delay and the freezing temperature depression of sessile water droplets (T~0 °C), respectively, are analyzed in-situ in a home-made environmental chamber, shown in pictorial view elsewhere [21]. Concisely, a perpendicular cylindrical copper rod, soldered to the copper base of a plexiglass chamber, is employed to transfer heat energy from a tank, with liquid nitrogen to the cryogenic system. Since the ambient temperature and humidity in the laboratory are relatively constant (*T_amb_*~22 ± 1 °C, *RH*~25 ± 4%), the evaporation rate of the liquid nitrogen is registered to be steady, allowing well-controlled cooling if the needed amount of coolant is added in a timely manner. A transparent lid, whose roof has two windows for dispensing the droplets, is mounted on the base and used to isolate the liquid loaded QCMs from externalities (e.g., dust particles, spontaneous condensation, etc.). The undesired temperature gradients are further minimized due to the high thermal conductivity of the copper base and copper crystal holder, and by setting the sensors next to the 2K-type probe of a Signstek 3^1/2^ 6802 II dual channel digital thermometer (Amazon, Bellevue, DC, USA), supporting thermal measurements with an accuracy of ±0.1–0.4 °C.

Each soot coated QCM (see Section 2.1) was mounted one at a time in the copper holder and firmly attached to the base. Once the resonance response in air was stabilized (i.e., Δ*f*~±1 Hz/s), the copper rod was immersed in liquid nitrogen and the chamber was cooled from room temperature to −30 °C for the case of bulk nucleation analysis. Then, a 50 μL water droplet was gently fixed in the middle of the QCM’s surface and its solidification and subsequent thawing at room temperature were contemporaneously tracked by means of the quartz sensor and a photo camera. The contour ice nucleation study was performed in the same way, but the droplet was placed on the soot coated QCM before initiating the cooling, executed with a rate of ~3.5 °C/min (identical to the rate obtained in other experiments [31,35]). Both types of tests were repeated in three independent measurement cycles in order to verify the degree of reproducibility of the sensor signal.

## 3. Results and Discussion

### 3.1. Detecting Bulk and Contour Ice Nucleation Modes through Soot Coated QCMs

Figure 3a,b depict the occurrence of bulk and contour freezing in 50 μL water droplets when evaluating their freezing time delay and freezing temperature depression on super-nonwettable soot coatings.

As seen, upon simultaneous cooling of the soot coating and the droplet, the recalescent freezing encompasses solidification of the liquid’s outer shell (Figure 3b), which appears to be a modification of the spectacular outside-in freezing, where the incipiency of ice crystals is triggered solely at the liquid–air interface [19,20]. This intriguing phenomenon is further investigated with soot coated QCMs, and the outcome is shown in Figure 4, where the main hypothesis is that the differences in the mass loading during the liquid–solid phase transitions will enlighten more on the underlying physics of the bulk and contour icing.

The careful examination of both resonance curves unveils several prominent characteristics of the recorded signals. First, the droplet’s solidification leads to ceased oscillations and the QCMs get out of resonance (*f*~1 MHz; *R*~5000 Ω), similarly to what is observed when a frost layer densifies on a soot coating (see Figure 3 in ref. [31]). Second, the thawing stage is accompanied by a restoration of the sensor response, but one curious result is clearly visible as well. Namely, after full liquefaction of the droplet, the value of the series resistance *R* at contour freezing is closer to the baseline compared to the case of bulk nucleation (see Figure 4a,b). In addition, at equal thawing rates (governed by the constant room temperature), the QCM registering the bulk icing seems to need a longer time to reach the frequency baseline (~1600 s in Figure 4a and ~1200 s in Figure 4b). These observations hint at very distinct ice adhesion forces supported by both freezing modes. Still, the graphs presented in Figure 4 do not provide information about the performance of the super-nonwettable QCMs at the moment of initial freezing, which is relevant to clarifying the physics of ice incipiency.

Figure 5 attempts to circumvent the foregoing limitations and yields the resonance behavior immediately prior to and after the initial droplet freezing.

The left side of Figure 5 demonstrates the temperature-induced sensor response that is generated by the thermal expansion/compression of the quartz crystal, directly affecting the vibration frequency [29,36,37,38]. Expectedly, the rapid thermal fluctuations inflict positive frequency shifts if the surface is not preliminary loaded with a water droplet, in perfect agreement with the previously documented dynamic temperature-frequency characteristics of 16 MHz soot coated QCMs (see Figure 2 in reference [31]). In contrast, the liquid load contributes to a more complex resonance behavior and it seems that the droplet’s viscosity/density alterations cause reciprocal downward drift of the frequency at some point, in compliance with the model proposed to account for the hybrid temperature effect [38].

The as-prepared QCMs react differently to the bulk and contour freezing and, as demonstrated in Figure 5a, a few seconds following the droplet deposition, the initial freezing drives the sensor out of resonance. Instead, the solidification of the droplet’s shell does not instantly harm the resonance and the sooted crystal continues to maintain oscillations for a certain time. Moreover, during the contour nucleation, the resonance frequency increases (see Figure 5b), which is surprising if one considers the operation principles of the QCM [22,23]. All of the described tendencies are reproducible from measurement-to-measurement, unequivocally reaffirming their physical origin (see Appendix A).

Although useful and exciting, the real-time graphical dependencies cannot be used to extract an additional information concerning the exact elapsed time before the QCMs get out of resonance, nor the precise values of the frequency, resistance and displaced mass shifts upon recalescent freezing, initial and full thawing (the numerical values at full freezing are not available, since the latter is preceded by out-of-resonance events). Therefore, we analyzed in detail the automatically saved numerical data sheets (see the Excel file in the SI), highlighted each milestone in the file in red color, and then summarized the data in Table 1, Table 2 and Table 3.

According to the numerical data, the beginning of the bulk freezing at −30 °C triggers frequency downshifts complemented by series resistance and displaced mass upshifts (see Table 1 and Table 2)—a result perfectly reflecting the physical basis of the Sauerbrey’s equation [22]. Interestingly, the complete liquefaction of the icy droplet (T~17 °C) moves the signal above the baseline i.e., an upward frequency deviation is noted (see the data of Δ*f_FT_* in Table 1). Inversely, the contour mode stimulates steady frequency upshifts during the initial freezing stage (T~−14 °C), while the full thawing (T~12 °C) is marked by displacement of the resonance frequency below the baseline (see the values of Δ*f_FT_* in Table 1). Furthermore, juxtaposing the trend of the series resistance for both nucleation modes shows that upon thawing, the value of Δ*R* at bulk icing is up to 9 times larger than at contour freezing (see the data of Δ*R_FT_* in Table 1 and Table 2). Last but not least, the out-of-resonance event is delayed, approximately, by a factor of 100 when the nucleation is performed simultaneously at the contact interface/droplet’s shell, and in all three experiments, the icy droplet liquefies faster (see Table 3). The gathered experimental findings imply intricate ice formation mechanisms on the soot coatings, which can be correlated to the generated QCM signals.

### 3.2. Insights into the Physical Origin of the Sensor Responses

When the QCM sensor operates in air, the strong acoustic reflectivity at the substrate–air interface localizes the wave energy in the crystal itself, whose volume (thickness) plays the role of a resonance cavity in which a standing wave is created [39]. As soon as the QCM is loaded with water, a viscous entrainment of the latter occurs and part of the wave energy dissipates in the liquid, which is expressed quantitatively by the following two equations [23,40]:(1)$$\Delta f=-f^{\frac{3}{2}}\sqrt{\frac{\rho \eta }{\pi \mu_{q}\rho_{q}}}$$
(2)$$\Delta R=\left(\frac{n\omega L}{\pi }\right).\left(\sqrt{\frac{2\omega \rho \eta }{{µ}_{q}\;\rho_{q}}}\right)$$
where *ρη* is the viscosity–density product of the liquid, *πµ_q_ρ_q_* denotes the electromechanical (acoustic) impedance of the quartz crystal, *n*—number of sides in contact with the liquid, *ω*—angular frequency at series resonance, and *L*—inductance at unloaded (dry) sensor.

If part of the water freezes, the solid–liquid sensing interface is replaced by a solid–solid (ice) one, creating strong quartz-ice elastic coupling [32], extension of the resonance cavity (within the ice layer’s thickness) [39] and a corresponding frequency decrease, according to the Sauerbrey’s equation [22]:(3)$$\Delta f=\frac{-2f^{2}}{A\sqrt{\mu_{q}\rho_{q}}}\Delta m$$
where *f* is the resonance frequency (Hz); Δ*f*—frequency shifts (Hz); *A*—active electrode area (cm^2^); Δ*m*—mass changes upon binding events (g). However, Equation (3) is valid for film thicknesses up to 1 μm [41] and mass loadings not exceeding 2% of the crystal’s mass [42]. Beyond that range, nonlinear effects take place and the frequency does not necessarily decrease [43].

Bearing in mind the theoretical concepts described above, we interpret the recorded sensor responses in light of the mass and viscosity–density changes at the interface. In case of bulk freezing (observed by analyzing the water’s ice nucleation-lag), the droplet deposition on the cooled soot coated QCM triggers wave penetration in the liquid and frequency downshifts (resistance upshifts) proportionally to the viscosity and density of the water (Equations (1) and (2)). Since the non-wettable surfaces delay the liquid–solid phase transitions by reducing the contact area and the heat transfer rate [44,45], the recalescent freezing does not happen immediately and the sensor generates continuous oscillations until promptly getting out of resonance once the interface solidifies (see Table 3 and the Appendix A). Based on the data in Table 2 (e.g., Δ*m_IF_* − Δ*m_TFC_* = 25 μg/cm^2^), and knowing that the density of ice is ~0.92 g/cm^3^, the ice layer’s thickness at that particular moment is calculated as ~273–556 nm (1st–3rd experiment). On the other hand, the unexpected positive frequency and negative mass shifts during the contour freezing may have two explanations: As stated elsewhere, “*the abnormally large increase in the resonance frequency upon the freezing of the water droplet is attributed to changes in the surface stress of the quartz crystal substrate. The increase in the contact radius during the freezing of the water droplet induces an increase in the surface stress of the quartz crystal, which increases the resonance frequency*
*and decreases the*
*Q**-factor of the quartz crystal*” [32]. While such a hypothesis is very reasonable and sound, we feel that the frequency increase is caused by instantaneous freezing of the contact interface and droplet’s contour, engendering partial delamination of the soot coating; because, in this study, we use pristine (non-chemically modified) quasisquare-shaped soot that is robust under water, but not resistive to mechanical interventions (abrasion, solid–solid friction, etc.) [33]. In turn, taking away a material from the sensing surface leads to shortening of the resonance cavity and displacement of the resonance to higher frequencies. Such a statement is upheld by the appearance of soot particles in the liquefied water droplet. Meanwhile, the enlargement of the liquid’s contact area upon freezing enhances the mechanical stress on the surface, which deteriorates the quality factor expressed here via the series resistance values. In addition, the smaller upward drift of *R* towards the baseline, over the droplet’s liquefaction, at the contour mode compared to the bulk analogue (see the values of Δ*R_FT_*−Δ*R_TFC_* in Table 2) indicates lower wave energy losses as a result of weaker liquid–solid contact (adhesion) [39]. Although speculative, this allegation is more or less validated by the shorter time required for complete droplet liquefaction at the contour freezing collated to the bulk freezing (see Table 3). The ice adhesion strength on solid surfaces is regulated by the surface temperature and the liquid-solid contact area – at constant negative temperature, larger the interfacial contact, stronger the ice adhesion and higher the thermal energy and/or longer the heating process to disrupt the ice bonds, and vice versa [46,47]. Hence, the elapsed time prior to resuming the resonance oscillations of the soot coated QCM is directly correlated to the ice adhesion forces.

Finally, the sensor signal is reproducible from measurement-to-measurement (i.e., three identical soot coated QCMs generate similar responses to a specific stimulus [48]), but not repeatable (i.e., the nominally equal sensors do not induce the same signal with regard to quantitative values [48]). One of the reasons could be related to the ice layer’s thickness, found to vary within the different experiments likely due to the stochastic genesis of the heterogeneous nucleation. Another explanation is that albeit equal in thickness, the carbon soot coatings may support divergent stress relaxation routes during the cooling process (e.g., at the electrode-coating interface), directly affecting the resonance behavior [49]. This would be a serious drawback for the proper practical utilization of our method, but seems unlikely, because at relatively equal Δ*m_TFC_* ~−78 ÷ (−81) μg/cm^2^ or Δ*m_TFC_* ~1.4÷1.5 μg/cm^2^, the displaced mass upon initial freezing (Δ*m_IF_*) differs up to a factor of 4 (see Table 1). In other words, it is very possible that the arbitrariness of the ice formation processes is one of the principal reasons for having unrepeatable readings.

## 4. Conclusions

The first systematic analysis of the ice nucleation dynamics of sessile water droplets by means of soot coated QCMs revealed that the formation of ice embryos prevalently at the liquid–solid interface instigated consistent resonance frequency decrease and series resistance increase, in conformity with the mass loading effects theoretically predicted by the Sauerbrey’s equation. The compact interfacial icing ceased the continuous oscillations immediately and the non-wettable piezoresonance devices got out of resonance until partial droplet liquefaction occurred. Excitingly, when the heterogeneous ice nucleation was driven jointly at the three-phase contact line and droplet’s contour, the QCM sensors were capable of vibrating at resonance for about 60–150 s. Moreover, unanticipated frequency increase was regularly identified during the contour freezing, complemented by faster recovery of the resonance upon room temperature thawing of the icy droplet. The collected experimental results suggested for different ice adhesion forces maintained by the bulk and contour freezing modes, which is of potential relevance to explicating (in the future) the physical processes governing the nascency of ice nuclei on solid particles or functional coatings. The complexity of this task could be alleviated by using the QCM technology, whose efficiency was undisputedly demonstrated herein.

## 5. Patents

A Bulgarian patent application (No. 113331) was submitted on 26 February 2021 as a result of the reported work.

## Figures and Tables

**Figure 1 micromachines-12-00463-f001:**
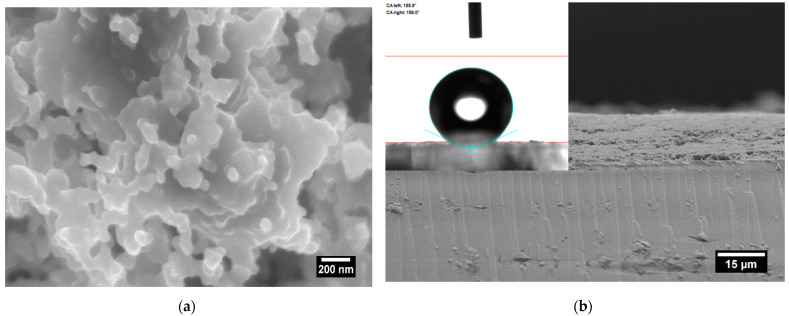
Top-view and cross-sectional scanning electron micrographs indicating (**a**) the morphology and structure, and (**b**) the thickness of the soot coatings, where the substrate-soot interface is easily distinguishable. The inset in Figure 1b reveals the static contact angle of a 10 μL water droplet resting on the soot.

**Figure 2 micromachines-12-00463-f002:**
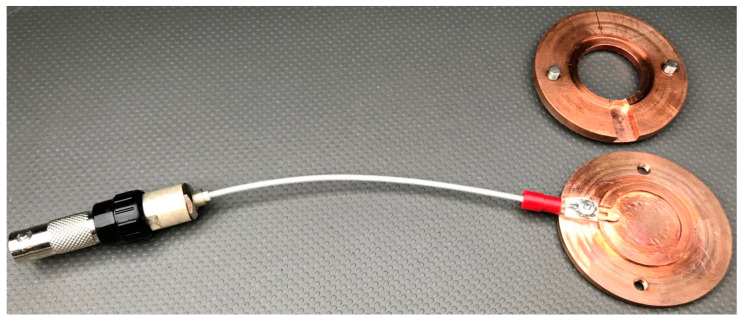
A custom copper-based QCM holder developed in “Acoustoelectronics” Laboratory at ISSP-BAS.

**Figure 3 micromachines-12-00463-f003:**
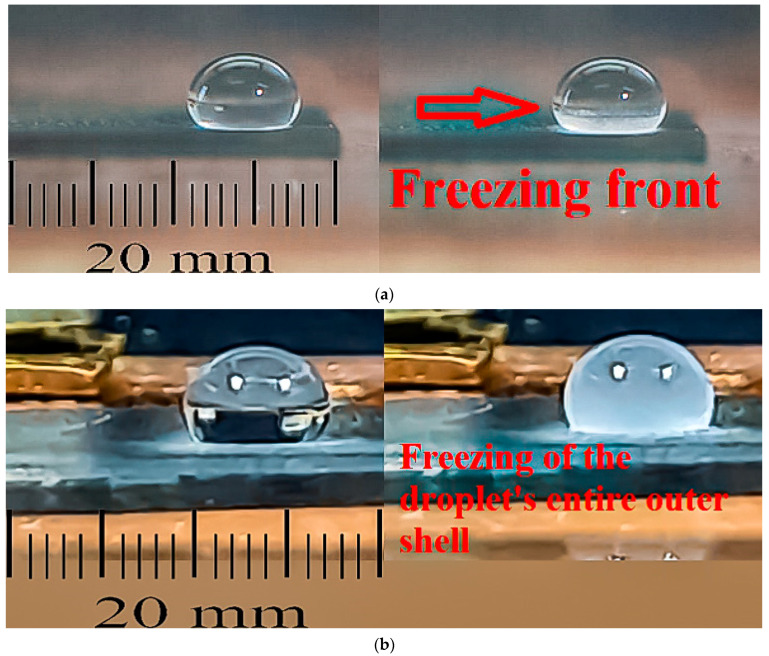
Photographs of (**a**) bulk and (**b**) contour freezing of 50 μL water droplets resting on super-nonwettable carbon soot coatings.

**Figure 4 micromachines-12-00463-f004:**
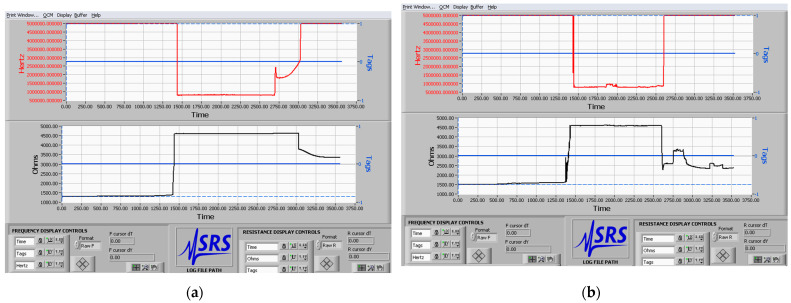
Real-time sensor response of two 5 MHz soot coated QCMs triggered by the (**a**) bulk and (**b**) contour freezing of a 50 μL water droplet placed in the middle of the sensing surface. The graphs are directly snapped from the PC’s screen instead of being processed by a specific software (e.g., OriginLab), with an aim of proving the authenticity of our experiments.

**Figure 5 micromachines-12-00463-f005:**
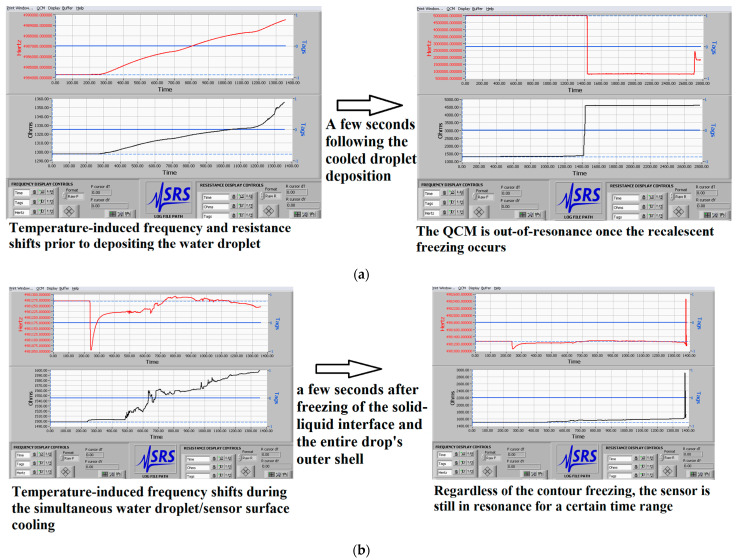
Real-time sensor response of two 5 MHz soot coated QCMs immediately prior to (left side) and after the initial solidification (right side) of a 50 μL water droplet placed in the middle of the sensing surface during (**a**) bulk and (**b**) contour freezing. The graphs are directly snapped from the PC’s screen instead of being processed by a specific software (e.g., OriginLab) with an aim of proving the authenticity of our experiments.

**Table 1 micromachines-12-00463-t001:** Raw data extracted from the automatically saved numerical data sheets. **Δ*f_TFC_***; **Δ*R_TFC_***; **Δ*m_TFC_*** denote the temperature-induced resonance frequency, series resistance and mass displacement shifts, respectively. **Δ*f_IF_***; **Δ*R_IF_***; **Δ*m_IF_*** represent the resonance frequency, series resistance and mass displacement shifts upon initial droplet freezing (i.e., recalescent freezing). **Δ*f_IT_***; **Δ*R_IT_***; **Δ*m_IT_*** indicate the resonance frequency, series resistance and mass displacement shifts upon initial droplet thawing. **Δ*f_FT_***; **Δ*R_FT_***; **Δ*m_FT_*** mark the resonance frequency, series resistance and mass displacement shifts upon complete droplet thawing.

Freezing mode	Δ*f_TFC_* (Hz)	Δ*f_IF_* (Hz)	Δ*f_IT_* (Hz)	Δ*f_FT_* (Hz)	Δ*R_TFC_* (Ω)	Δ*R_IF_* (Ω)	Δ*R_IT_* (Ω)	Δ*R_FT_* (Ω)	Δ*m_TFC_* (μg/cm^2^)	Δ*m_IF_* (μg/cm^2^)	Δ*m_IT_* (μg/cm^2^)	Δ*m_FT_* (μg/cm^2^)
bulk	5587	3110	636	285	81	1482	2693	2039	−99	−55	−11	−5
	4582	3180	4305	413	−125	531	2405	905	−81	−56	−76	−7
	4427	1522	2627	294	3	1238	2753	1852	−78	−27	−46	−5
contour	−86	750	5197	−393	164	384	2435	869	1.5	−13	−92	7
	−366	4644	4922	−380	396	844	2671	803	6.5	−82	−87	7
	−77	3235	5619	−308	50	935	2027	237	1.4	−57	−99	5.4

**Table 2 micromachines-12-00463-t002:** Actual sensor response reflecting the water droplet’s freezing and thawing (after subtracting the temperature effects).

Freezing Mode	Δ*f_IF_*−Δ*f_TFC_* (Hz)	Δ*f_IT_*−Δ*f_TFC_* (Hz)	Δ*f_FT_*−Δ*f_TFC_* (Hz)	Δ*R_IF_*−Δ*R_TFC_* (Ω)	Δ*R_IT_*−Δ*R_TFC_* (Ω)	Δ*R_FT_*−Δ*R_TFC_* (Ω)	Δ*m_IF_*−Δ*m_TFC_* (μg/cm^2^)	Δ*m_IT_*−Δ*m_TFC_* (μg/cm^2^)	Δ*m_FT_*−Δ*m_TFC_* (μg/cm^2^)
bulk	−2477	−4951	−5302	1401	2612	1958	44	88	94
	−1402	−277	−4169	656	2530	1030	25	5	74
	−2905	−1800	−4133	1235	2750	1849	51	32	73
contour	836	5283	−307	220	2271	705	−14.5	−93.5	5.5
	5010	5288	−14	448	2275	407	−88.5	−93.5	1.5
	3312	5696	−231	885	1977	187	−58.4	−100.4	4

**Table 3 micromachines-12-00463-t003:** Numerical data accounting the time needed for recalescent freezing (***t_IF_***), out-of-resonance events (***t_OR_***), initial (***t_IT_***), and full thawing (***t_FT_***).

Freezing Mode	*t_IF_* (s)	*t_OR_* (s)	*t_IT_* (s)	*t_FT_* (s)	*t_IT_ + t_FT_ (s)*
bulk	12	1	1598	859	2457
	24	1	1090	1217	2307
	7	2	978	935	1913
contour	n/a	62	1167	1010	2177
	n/a	102	1149	1105	2254
	n/a	149	866	643	1509

## Data Availability

The data presented in this study are available on request from the corresponding author.

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
