# Peer review of "Studying the Bulk and Contour Ice Nucleation of Water Droplets via Quartz Crystal Microbalances"

_micromachines, 2021, doi:10.3390/mi12040463_

Round 1

Reviewer 1 Report

Dear authors,

Thank you for submitting this manuscript. Please see my detailed comments in the attachment. 

All the best!

Author Response

The authors would like to sincerely thank the reviewers for their time to review the submitted manuscript and for the provision of helpful and constructive comments, suggestions and recommendations. We did our very best to address them in the revised version of the manuscript. Revisions in the text related to the reviewers’ comments are highlighted in yellow. The responses are given below, each comment is shown in italic and the response shown in standard font.

Referee 1:

General comments

This manuscript induced a novel sensor system based on QCM to study the mechanism of water freezing, which is very inspiring and important in this field. Using this system, a new hybrid version of the outside-in freezing was discovered, in which the freezing was initiated both at the contact interface and droplet’s entire outer shell. Generally, the work was well organized and the results and discussions were well presented in the manuscript. Some improvements can be made in figures and more in-depth discussion. Overall, I think the manuscript can be published after minor revision.

Response:

We thank the reviewer for his/her positive evaluation of our manuscript and we are glad he/she considers the proposed research as inspiring and important. Undoubtedly, the manuscript’s quality can be improved based on all of the feedback received so far and we did our very best to satisfy the reviewers and editors expectations.

1) Page 4 figure2: The copper holder is really cool. Was a calibration performed with the copper holder regarding the temperature gradient or measuring an item with a known weight?

Response:

Thank you so much, we also believe that the copper holder has a nice design and possesses several advantages, described in the paper, compared to the commercially available QCM holders. In fact, we have submitted a patent application concerning the holder. And yes, prior to using it, we verified whether a mismatch between the copper base’s temperature and the holder’s temperature exist by using a thermal imaging camera FLIR P640. The maximum thermal deviation was found to be ±0.2 °C i.e., virtually no gradients are observed.

Change in Manuscript:

A sentence addressing the reviewer’s question is now available on Page 4 Lines 144-145 in the revised manuscript.

2) Page 6 figure 4 and 5: The resolution of figure 5 needs to be increased. Does the left image of figure 5a corresponds to figure 4a when time < 1400 s? Why does the resistance curve in the left image of figure 5a decrease with time but the resistance curve in figure 4a seems to increase with time?

Response:

Sincerely thank you for finding out this major shortcoming. You are absolutely correct that in Figure 4a the resistance increases with time (prior to the icing). We have done a mistake and Figure 5a does not correspond to Figure 4a prior to the 1400 s. Figure 4a was taken from the third measurement of the bulk nucleation, while Figure 5a reflects the first trial. This mistake is now fixed.

Change in Manuscript:

Figure 5a is now replaced with a new figure containing the correct graphical dependencies showing the moment immediately prior to and after the recalescent freezing, extracted from the overall response shown in Figure 4a. The new figure is available on Page 6 in the revised manuscript.

3) Page 8 table 1,2,3: It seems that the data of three independent measurements have large variations, even after subtracting temperature effects in Table 2 (e.g. ΔfIT − ΔfTFC for bulk freezing). Would you please discuss the possible reasons and the impact on the utilization of the method to study water freezing (e.g. the impact on ice thickness calculation)?

Response:

Another very relevant question, thanks. What we believe is the following: the sensor signal is reproducible from measurement-to-measurement (i.e., three identical soot coated QCMs generate similar response to a specific stimulus), but not repeatable (i.e., the nominally equal sensors do not induce the same signal with regard to quantitative values). One of the reasons could be related to the ice layer's thickness, found to vary within the different experiments likely due to the stochastic genesis of the heterogeneous nucleation. Another explanation is that albeit equal in thickness, the carbon soot coatings may support divergent stress relaxation routes during the cooling process (e.g., at the electrode-coating interface), directly affecting the resonance behavior. This would be a serious drawback for the proper practical utilization of our method, but seems unlikely, because at relatively equal ΔmTFC ~ -78÷-81 μg/cm2 or ΔmTFC ~1.4÷1.5 μg/cm2, the displaced mass upon initial freezing (ΔmIF) differs up to a factor of 4 (see Table 1). In other words, it is very possible that the arbitrariness of the ice formation processes is one of the principal reasons for having unrepeatable readings.

Change in Manuscript:

Discussion related to the possible reasons for the lack of repeatability of the sensor signal is now available on Page 10 Lines 325-337 in the revised manuscript. Another two references, number 48 and 49, are added in order to support our conclusions.

4) Page 3 figure 1b: The inset of the contact angle picture is too small. Is the thickness of soot coating shown in figure 1b? It is hard to tell if an interface exists at about 16 um.

Response:

Yes, figure 1b shows the soot film thickness. We agree that the inset is too small, therefore, it is enlarged now. Furthermore, we added a sentence stating that the interface on figure 1b is clearly visible, which is true if one inspects the image more carefully.

Change in Manuscript:

The inset in Figure 1b is now enlarged and a sentence addressing the reviewer’s concern is added in the caption of Figure 1 on Page 3 Line 115 in the revised manuscript.

5) Page 5: the droplets in figure 3b were not in focus.

Response:

We agree. The problem with the real-time imaging was that the droplets are placed inside the chamber i.e., covered by a Plexiglas lid. This lid deteriorates the quality of the video imaging and that is why sometimes the droplets were on focus, whereas in other cases the substrate was more focused than the droplets. We tried to process the snapshots using photoshop and in our opinion, now they look good.

Change in Manuscript:

Figure 3b is now replaced by a new figure with higher resolution, available on Page 5 in the revised manuscript.

Reviewer 2 Report

This work analyzed the ice nucleation dynamics of sessile water droplets, by means of soot coated QCMs, systematically for the first time. The new sensor configuration is proposed to distinguish whether the inception of ice nuclei is solely at the solid-liquid interface or along the droplet’s entire shell. They revealed that the formation of ice embryos prevalently at the liquid-solid interface instigated consistent resonance frequency decrease and series resistance increase, in conformity with the mass loading effects theoretically predicted by the Sauerbrey’s equation. In general, this whole manuscript tells a clear experimental and theoretical investigation, however, the minor revisions need addressing before publication:

  1. “These particles are deposited on the quartz crystals’ surface by subjecting the latter to the flame/fume emitted from the chimney’s tip for 15 s. ”  How can authors make sure that the soot coating is uniform no the substrate?
  2. Figure 2 and Figure 3 are too blurry. May you provide the photos of high quality? 
  3. Please add scale bars in graphs of Figure 3. 

Author Response

The authors would like to sincerely thank the reviewers for their time to review the submitted manuscript and for the provision of helpful and constructive comments, suggestions and recommendations. We did our very best to address them in the revised version of the manuscript. Revisions in the text related to the reviewers’ comments are highlighted in yellow. The responses are given below, each comment is shown in italic and the response shown in standard font.

Referee 2:

This work analyzed the ice nucleation dynamics of sessile water droplets, by means of soot coated QCMs, systematically for the first time. The new sensor configuration is proposed to distinguish whether the inception of ice nuclei is solely at the solid-liquid interface or along the droplet’s entire shell. They revealed that the formation of ice embryos prevalently at the liquid-solid interface instigated consistent resonance frequency decrease and series resistance increase, in conformity with the mass loading effects theoretically predicted by the Sauerbrey’s equation. In general, this whole manuscript tells a clear experimental and theoretical investigation, however, the minor revisions need addressing before publication:

1) “These particles are deposited on the quartz crystals’ surface by subjecting the latter to the flame/fume emitted from the chimney’s tip for 15 s.”  How can authors make sure that the soot coating is uniform no the substrate?

Response:

Thank you very much for supporting our research, we are glad that its content is interesting to the reviewer. We are sure that the coating is uniform due to the following reasons: First, the flame/fume is laminar i.e., the fluid flow is consistent. Second, the coating deposition takes place with fluent circular movement of the specimens across the fume. Third, the film’s uniformity has been verified previously (please see Table 1 in https://www.sciencedirect.com/science/article/pii/S0927775719300482), as well as during this study. The maximum thickness deviation is ±1 μm.

Change in Manuscript:

A sentence addressing the reviewer’s concern is now available on Page 3 Lines 108-109 in the revised manuscript.

2) Figure 2 and Figure 3 are too blurry. May you provide the photos of high quality?

Response:

Absolutely agree, thanks.

Change in Manuscript:

Figures 2 and 3 are now replaced by figures with much higher quality.

3) Please add scale bars in graphs of Figure 3.

Response:

Thanks for this precious advice. It has been taken into account.

Change in Manuscript:

Scale bars are now available at the lower end of the images presented in Figure 3.

Reviewer 3 Report

The paper is devoted to the experimental study of water crystallization using the quartz crystal microbalance technique. The experimental procedure is given in detail. The bulk and surface ice formation are observed. A few concerns should be addressed.

  1. Abstract. “Due to the stochastic and time-dependent nature of the ice embryo formation and growth”. The phrase “stochastic nature of growth” is unclear and should be reformulated.
  2. Is the term “freezing time delay” equivalent to the nucleation time-lag? It should be clarified.
  3. At which temperature the data shown in Tables 1 & 2 were obtained?

In my opinion, the manuscript can be published in the Micromachines after revision and English language polishing.

Author Response

The authors would like to sincerely thank the reviewers for their time to review the submitted manuscript and for the provision of helpful and constructive comments, suggestions and recommendations. We did our very best to address them in the revised version of the manuscript. Revisions in the text related to the reviewers’ comments are highlighted in yellow. The responses are given below, each comment is shown in italic and the response shown in standard font. 

Referee 3

The paper is devoted to the experimental study of water crystallization using the quartz crystal microbalance technique. The experimental procedure is given in detail. The bulk and surface ice formation are observed. A few concerns should be addressed.

1) Abstract. “Due to the stochastic and time-dependent nature of the ice embryo formation and growth”. The phrase “stochastic nature of growth” is unclear and should be reformulated.

Response:

We are very grateful to the reviewer for his/her positive assessment of the manuscript. The heterogeneous nucleation is stochastic by nature, since it has random distribution. It can be analyzed statistically, but cannot be predicted precisely. Please see Vali, G. Interpretation of freezing nucleation experiments: singular and stochastic; sites and surfaces. Atmos. Chem. Phys. 2014, 14, 5271-5294, doi:10.5194/acp-14-5271-2014. Wright, T. P.; Petters, M. D. The role of time in heterogeneous freezing nucleation. J. Geophys. Res. Atmos. 2013, 118, 3731-3743, doi:10.1002/jgrd.50365.

Change in Manuscript:

A sentence addressing the reviewer’s concern is available in the abstract of the revised manuscript.

2) Is the term “freezing time delay” equivalent to the nucleation time-lag? It should be clarified.

Response:

This is an important question, thanks. Yes, in our opinion, “freezing time delay” is equivalent to “nucleation time-lag”.

Change in Manuscript:

A phrase addressing the reviewer’s question is now available on Page 2 Line 70 in the revised manuscript.

3) At which temperature the data shown in Tables 1 & 2 were obtained?

Response:

Thank you for this question. The data in Tables 1-2 dealing with the bulk nucleation are obtained at -30 °C, for the freezing process, and around +17 °C substrate/holder temperature for the complete thawing. In contrast, the data for the contour nucleation are obtained at -14 °C, during the initial freezing, and at +12 °C for the complete thawing.

Change in Manuscript:

Clarification regarding the substrate/holder temperatures at which the data in Tables 1-2 are obtained is now available on Page 7 Lines 239-246 in the revised manuscript.

4) In my opinion, the manuscript can be published in the Micromachines after revision and English language polishing.

Response:

Thank you very much indeed for recommending acceptance of our paper. We firmly believe that the level of the English language meets the high standards of Micromachines and other reputable journals, as kindly noted by the other two reviewers.

Change in Manuscript:

We have paid full attention to the English language and applied the required stylistic modifications where needed.